# *DOCK8* Mutation in Patient with Juvenile Idiopathic Arthritis and Sjögren’s Syndrome

**DOI:** 10.3390/ijms25042259

**Published:** 2024-02-13

**Authors:** Violetta Opoka-Winiarska, Natalia Winiarska, Monika Lejman, Małgorzata Gdak, Krzysztof Gosik, Filip Lewandowski, Paulina Niedźwiedzka-Rystwej, Ewelina Grywalska

**Affiliations:** 1Department of Paediatric Pulmonology and Rheumatology, Medical University of Lublin, 20-093 Lublin, Poland; violetta.opoka-winiarska@umlub.pl; 2Department of Methodology, Medical University of Warsaw, 02-091 Warsaw, Poland; natalia.winiarska@hotmail.com; 3Laboratory of Genetic Diagnostics, Medical University of Lublin, 20-093 Lublin, Poland; monika.lejman@umlub.pl; 4University Children’s Hospital in Lublin, 20-093 Lublin, Poland; malgorzata.gdak@gmail.com; 5Department of Experimental Immunology, Medical University of Lublin, 20-093 Lublin, Poland; krzysztof.gosik@umlub.pl (K.G.); ewelina.grywalska@umlub.pl (E.G.); 6Institute of Biology, University of Szczecin, 71-412 Szczecin, Poland; 227077@stud.usz.edu.pl

**Keywords:** *DOCK8*, juvenile idiopathic arthritis, rheumatic diseases, Sjogren’s syndrome, autoimmunity

## Abstract

This study investigated the association between autoimmunity and immunodeficiency in pediatric patients, focusing on the case of a 15-year-old female diagnosed with juvenile idiopathic arthritis (JIA) and secondary Sjögren’s syndrome. The patient presented with a variety of symptoms, including joint pain, bronchial asthma, leukopenia, and skin lesions. Genetic testing revealed a de novo mutation in the *DOCK8* gene, associated with *DOCK8* deficiency, a condition usually associated with immunodeficiencies. The clinical course, diagnostic pathway, and treatment history are detailed, highlighting the importance of molecular diagnostics in understanding the genetic basis of rheumatic diseases. This case highlights the need to consider innate immune errors in patients with multiple diseases or atypical symptoms of rheumatic diseases. Furthermore, the study highlights the importance of targeted treatment, including genetic counseling, to improve patient outcomes. The observed association between autoimmunity and immune deficiency reinforces the importance of molecular testing in elucidating the causes of previously idiopathic rheumatic diseases, contributing to improved patient care and quality of life.

## 1. Introduction

Pediatric autoimmune diseases may pose diagnostic and therapeutic problems. Sjögren disease (SD) and juvenile idiopathic arthritis (JIA) are theorized to have a polygenic nature and respond to environmental factors. Inborn errors of immunity (IEIs) can manifest as autoimmune conditions [1,2]. According to the International Union of Immunological Societies (IUIS) expert committee (EC) on inborn errors of immunity (IEIs), in the 2022 updated phenotypic classification, *DOCK8* deficiency belongs to the group of immunodeficiencies affecting cellular and humoral immunity (combined immunodeficiencies with low B cells) [3]. The *DOCK8* gene contains 48 exons and is located on the short arm of the chromosome locus 9p24.3. This gene encodes a member of the DOCK180 family of guanine nucleotide exchange factors. Guanine nucleotide exchange factors interact with Rho GTPases and are components of intracellular signaling networks. Mutations in this gene result in the autosomal recessive form of the hyper-IgE syndrome [1]. *DOCK8* is most abundant in cells of the immune system and plays a key role in the survival and function of several types of immune cells, including T cells, natural killer (NK) cells, and B cells. It is also involved in chemical signaling pathways that stimulate B cells to mature and produce antibodies. *DOCK8* expression has been found in the placenta, lung, pancreas, and kidney [4]. In 2009, Zhang et al. [5] first identified deletions or mutations in the *DOCK8* gene in both homozygous and heterozygous systems. Patients had recurrent and severe bacterial, viral, and fungal infections, atopic dermatitis, and food and environmental allergies. A few developed virus-related cancers. Laboratory tests showed decreased T lymphocytes, a variable decrease in B lymphocytes, and elevated serum IgE levels. Since then, there have been attempts to further investigate the exact role of *DOCK8* in the immune system and possible manifestations of occurring mutations [5,6,7,8]. Although mostly associated with immunodeficiencies and allergies, *DOCK8* deficiency was also associated with autoimmune diseases such as atopy, hemolytic anemia, cytopenia, vasculitis, uveitis, and systemic lupus erythematosus (SLE) [1,5,7,9]. The immunological effects of *DOCK8* deficiency are summarized in Table 1. We present a novel presentation of *DOCK8* mutation in a patient with a primary diagnosis of JIA and SD.

## 2. Case Description

A *DOCK8* gene mutation was detected in a 15-year-old female patient diagnosed with juvenile idiopathic arthritis and secondary Sjogren’s syndrome, bronchial asthma, and leukopenia. The patient was vaccinated according to the national vaccination program, with no post-vaccination reactions.

### 2.1. Clinical Manifestations

The first manifestation occurred at the age of 7 years. The patient complained of increasing joint pain in the upper and lower limbs and morning stiffness. The results of the baseline investigations were normal, and treatment was managed by the family doctor. The child’s development was normal, and the course of the infection was typical of childhood.

At 10 years of age, the patient was referred to a rheumatology consultation with symptoms of polyarthritis. Physical examination showed signs of arthritis in the upper and lower limbs with significant limitation of mobility.

At the age of 11 years, during a routine follow-up visit, a blood count showed leukopenia, neutropenia, and anemia. Her previous DMARD therapy (sulfasalazine and methotrexate) was discontinued due to suspected treatment-related side effects, but her blood cell count was still in the lower limit of normal. She was consulted by a hematologist for leukopenia and neutropenia; despite the termination of treatment, the myelogram was normal, no anti-neutrophil antibodies were found, and the blood lymphocyte subpopulation had a below-normal NK cell count.

At the age of 13, the patient developed symptoms of bronchial asthma and was diagnosed with skin lesions in the form of acanthosis nigricans and pityriasis rosea.

The patient had normal values of the B-lymphocyte subpopulation (CD19+), while the number of T lymphocytes (CD3+) was slightly elevated. A significantly lower number of NK cells (CD16+, CD56+) was observed than in healthy subjects. The number of helper T cells (Th; CD3+/CD4+) was also increased compared to normal. Typically, the number of helper T cells is inversely correlated with a decreased titer of cytotoxic T cells (Tc; CD3+/CD8+) compared to normal. Importantly, a low proportion of CD25+ cells was observed in both B- and T-lymphocyte populations. These cells, both B and T lymphocytes, show maturity and antimicrobial activity. This may indicate an acquisition disorder. The exceptionally high expression of programmed cell death factor 1, a protein complex that activates apoptosis in effector immune cells expressing the PD-L1 ligand, may be related to the low number of activated CD25 cells.

### 2.2. Diagnostic Pathway

Laboratory investigations at 10 years of age of the patient for signs of polyarthritis showed blood counts and leukocyte and neutrophil counts in the lower limit of normal, and inflammatory markers (erythrocyte sedimentation rate (ESR) and c-reactive protein (CRP)) were normal. The antinuclear antibody (ANA) titer was 1:320 and had a homogeneous pattern, and the extractable nuclear antigen (ENA) panel did not detect antibodies. Results for rheumatoid factor (RF) and anti-citrullinated protein antibodies (ACPAs) were negative, as was the HLA-B27 antigen result. C3 and C4 complement concentrations and immunoglobulins IgG, IgA, IgM, and IgE were normal. The myelogram was also normal. Imaging studies showed periarticular osteoporosis on a joint X-ray and ultrasound features of arthritis. Other causes of arthritis were ruled out, and the patient was diagnosed with seronegative polyarticular juvenile idiopathic arthritis (JIA) with high activity. The patient’s juvenile arthritis disease activity score of 71 (JADAS71) was 62. Viral infections, including EBV, CMV, HSV, and parvovirus B19, were ruled out during a routine visit at the age of 11 years. Complement and immunoglobulin levels (IgG, IgM, IgA, IgE) were normal, the ANA titer was 1:320, and the ENA test was negative.

A chest X-ray and CT scan performed at 13 years of age showed no abnormalities.

An ultrasound examination at 14 years of age showed parotid glands with normal echogenicity, slightly enlarged with numerous intraglandular nodes, larger and more numerous on the left side—up to 4 mm on the short axis. Vascularization by color Doppler was normal. The surrounding lymph nodes were not enlarged. A biopsy of the labial salivary gland showed inflammatory foci with lymphocyte infiltration, typical of Sjögren’s syndrome. The ophthalmological consultation showed no abnormalities—Schirmer’s test result above 20 mm, fundus normal.

Based on the clinical picture of the disease, taking into account the effect of immunosuppression after symptom management, a diagnosis of secondary juvenile Sjogren’s syndrome was made. The disease activity according to the ESSDAI (EULAR Sjögren’s disease activity index) was 10 points. IgG4-related disease without a diagnosis of this syndrome was included in the differential. Symptoms of salivary gland inflammation resolved after the GC dose was increased.

At the age of 16 years, due to an atypical course of inflammation of the right knee joint, an MRI scan was performed, which showed signs of pigmented villonodular synovitis (PVNS). An arthroscopic synovectomy was performed, and a histopathological examination of the synovium confirmed the diagnosis. PVNS is classified as a giant cell tumor of the tendon sheath, a group of rare, typically benign joint tumors.

A summary of the patient’s disease course is shown in Appendix A, and the results of baseline lymphocyte subpopulations are shown in Appendix A. In 2022, due to the patient’s multimorbidity, the idiopathic nature of her symptoms, and the lack of expected response to treatment, a genetic consultation and molecular testing were planned.

Whole-exome sequence studies were performed. Libraries were prepared using the Agilent SureSelect VI Exome Kit and Illumina technology at an average gene coverage depth of 100×. A panel of 59 genes associated with primary immunodeficiencies and autoinflammatory diseases was used in the first screening, and a panel of 372 genes associated with primary immunodeficiencies and autoinflammatory diseases was used in the subsequent screening. The lesions responsible for the patient’s clinical presentation were excluded. Deletions of several exons in the *BANK1* gene were detected in one of the gene copies; the result required confirmation with met MLPA or aCGH. In order to achieve this, we used the CytoScan HD microarrays. No deletions or duplications were observed in the *BLANK1* gene. Other potentially pathogenic alterations were detected: NOD2 c.3019dupC. NOD2 is a member of the nucleotide-binding oligomerization domain (NOD)-like receptor family of PRRs. This protein is primarily expressed in the peripheral blood leukocytes. It plays a role in the immune response to intracellular bacterial lipopolysaccharides (LPSs) by recognizing the muramyl dipeptide (MDP) derived from them and activating the NFKB protein. Upon detection of tissue damage or microbial infection, pattern-recognition receptors (PRRs) initiate inflammatory responses. According to the ClinVar database, this variation is classified as a risk factor for the development of Yao syndrome, Blau syndrome, Crohn disease, and Inflammatory Bowel Disease 1 (Crohn disease) [10]. It is also known as a factor that increases the predisposition to various cancers. In this regard, genetic counseling is recommended, which can help understand the risk of inheriting cancer and suggest appropriate genetic tests. Genetic counseling can assist in determining the individual risk of developing cancer and taking appropriate preventive actions.

The most important finding was a de novo mutation in the *DOCK8* gene: NM_001193536.1:c.624G>A NM_001193536.1c.624G>A(p.=) (Figure 1).
NM_001193536.1:c.624G>A NM_001193536.1c.624G>A(p.=)

The notation NM_001193536.1:c.624G>A denotes a change in the DNA sequence at position 624 where a guanine (G) has been replaced by an adenine (A). The (p.=) typically indicates a synonymous substitution, meaning that the change in the DNA sequence does not result in an amino acid alteration in the protein.

The mutation has a previously unknown clinical significance. There is a moderate likelihood of splicing disruption—an additional donor site for splicing is created. The alteration is located in the first nucleotide of the exon. This mutation may be responsible for the patient’s clinical picture associated with *DOCK8* immunodeficiency syndrome or complex immunodeficiencies caused by *DOCK8* deficiency. It is inherited in an autosomal recessive manner. *DOCX8* deficiency was diagnosed on the basis of the clinical picture and test results. This type of genetic variant may require further investigation, for example, whole-genome sequencing, to determine its potential clinical significance and whether it affects the function of the resulting protein. All detected mutations are shown in Table 2.

### 2.3. Treatment History

From the age of 10, non-steroidal anti-inflammatory drugs (NSAIDs), a disease-modifying anti-rheumatic drug (DMARD), methotrexate, corticosteroids (GCs), and a second sulfasalazine DMARD were successively included in the treatment, resulting in a slow decline in disease activity. Remission was achieved during treatment (JADAS71 -0), and NSAIDs and GCs were discontinued. Due to increased disease activity following DMARD treatment in year 11 (arthritis of the upper and lower extremities, JADAS71-24), treatment with the TNF inhibitor adalimumab was initiated. A second dose of the drug was followed by reddening of the skin near the injection site, abdominal pain, and vomiting. An allergic reaction to the drug was noted, and this therapy was also discontinued. An adverse reaction was reported. Subsequently, further conventional DMARDs were included in the treatment: hydroxychloroquine and cyclosporine with GCs administered intrathecally and orally, with blood morphology and laboratory inflammatory markers within normal limits. Due to persistently high disease activity despite this treatment, a biological DMARD, the TNF inhibitor etanercept, was added with good tolerability. Treatment was continued, reducing the dose of GCs. After the diagnosis of bronchial asthma at the age of 13 years, combination inhaler treatment (GCs with a bronchodilator) was recommended. At 14 years of age, the patient continued to be treated with a low-dose GC (prednisone 7.5 mg) and cyclosporine. Treatment with etanercept was discontinued due to low disease activity. During treatment, painful swelling of the salivary glands occurred.

Following the diagnosis of the *DOCK8* deficiency mutation, anti-inflammatory treatment with a low-dose GC and cyclosporine is continued. Unfortunately, any attempt to reduce the dose of the GC (currently prednisone 10 mg) results in an exacerbation of clinical symptoms.

## 3. Discussion

Clinical manifestations of *DOCK 8* deficiency include a broad spectrum of symptoms associated with combined immunodeficiency usually present at a young age and can affect male and female patients of various ethnic backgrounds [4,9]. They can manifest as allergic reactions, decreased immunity against infections, as well as autoimmunity and malignancy [5,7,9,11], (Table 3). *DOCK8* deficiency is also linked with higher morbidity and mortality [5,6,7,8,9].

Most often, clinical cases of *DOCK8* deficiency manifest as primary immunodeficiency and allergic diseases [5]. We describe a case with a dominant autoimmune manifestation.

The first clinical manifestation in the described case was JIA complicated by secondary Sjogren’s syndrome after several years. Additionally, the patient was diagnosed with allergic diseases: bronchial asthma and an allergic reaction to the drug. Immune disorders were also manifested by leukopenia with neutropenia and a reduced number of natural killer (NK) cells.

The patient also developed a benign tenosynovial giant cell tumor in the form of PVNS.

This set of symptoms corresponds to the manifestation of impaired *DOCK8* function, a protein for which a de novo gene mutation was diagnosed in the described patient.

Approximately 230 cases of *DOCK8* deficiency were described until 2017. The analysis of these cases showed that patients with *DOCK8* deficiency may present significant autoimmune symptoms. These include autoimmune hemolytic anemia or other cytopenias, vasculitis, chorioretinitis or uveitis, hypothyroidism, and more [10]. Systemic lupus erythematosus (SLE) was reported in one patient, a 10-year-old girl, with purpuric and necrotic skin lesions; arthritis; glomerulonephritis; and the presence of antinuclear, anti-DNA, and antiphospholipid antibodies [11]. Our observation in our knowledge is the first case of JIA and Sjogren disease within the literature where the primary immune disease is genetically documented as *DOCK8* deficiency.

Inborn errors of immunity (IEIs) are defined as immunological disorders consequences of damaging germline variants in single genes resulting in variable susceptibility to infections, immune dysregulation, or malignancies [12,13]. The relationship between innate immune defects and rheumatic diseases is a current and interesting problem. Many rheumatic diseases, including JIA and Sjogren’s disease, are defined as idiopathic. The wide availability of genetic tests and the presented case indicate the need to search for the cause of rheumatic diseases among inborn errors of immunity. In our opinion, molecular diagnostics should be especially considered in patients with multimorbidity, a family history of autoimmune diseases, or an atypical course of rheumatic disease.

The importance of correctly defining the disease for the patient should be emphasized.

The identification of the genetic and pathophysiological basis of the disease led to targeted management including therapy [1]. The patient presented with leukopenia and neutropenia; therefore, drugs such as methotrexate or sulfasalazine for arthritis treatment should be avoided. However, the effectiveness of cyclosporine administered periodically with low doses of GCs was good. Currently, the patient is in good condition, and her current health problem is PVNS; however, if the condition is severe, allogeneic hematopoietic stem cell transplantation (HSCT) may be considered [14,15]. Commonly reported treatment with HSCT showed patients’ improvement in infectious and atopic complications [5,7,8,9,16,17].

*DOCK8* deficiency may be associated with severe infections. The patient was vaccinated in accordance with the national vaccination plan, including against COVID-19, with good tolerance and a sufficient serological response. However, in the case of infections, the patient requires special observation. Immune defects caused by the *DOCK8* mutation can affect malignancies related to HPV infections, EBV-connected lymphomas, and skin cancers [4,5,7]. Awareness of this process requires careful monitoring of patients and conducting additional screening.

The limitation of the study is the lack of whole-genome sequencing (WGS), which would enable the examination of non-coding sequences and determine the pathogenicity of the detected variant in the *DOCK8* gene.

Finally, defining the disease helps explain the problem to the patient and caregivers, clarify the cause, and plan the management together, including genetic counseling. Despite the importance of the problem, its understanding is usually associated with improving the quality of life of the patient and family.

## 4. Conclusions

Our observations confirm the link between autoimmunity and immunodeficiency. Autoimmune disease may be the main symptom of an inherited immune disorder. Molecular tests may explain the cause of some rheumatic diseases that were previously considered idiopathic.

## Figures and Tables

**Figure 1 ijms-25-02259-f001:**
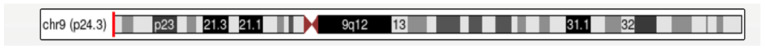
Gen *DOCK8*—dedicator of cytokinesis 8 (OMIM: 611432).

**Table 1 ijms-25-02259-t001:** Immunological effects of *DOCK8* deficiency.

Innate Immunity	Adaptive Immunity
↓ cell polarization and migration through 3-dimensional space	↓ regulation of the actin cytoskeleton, adhesion molecule accumulation, and immune synapse formation
↓ regulation of STAT3 phosphorylation, nuclear translocation, IL-22 production	T lymphocytes and NK cells undergo cytothripsis, and the generation of skin-resident memory CD8+ T cells is prevented
↓ function and survival of innate lymphoid cells (ILCs)	↓ B and T lymphocyte and NKT cell survival and long-lived memory responses
↓ plasmacytoid dendritic cells in circulation	↓ antibody production and generation of memory B cells; IgG, IgA, IgM, ↑, normal, or ↓
NK cells: defects in natural cytotoxicity	↓ Treg suppressive function and frequency
	NKT cells: defects in specific activating receptor-mediated cytotoxicity
	higher proportion of activated cells producing Th2 cytokines with eosinophilia and elevated IgE
	↓ naïve CD4+ T cells with ↓ Th17 cell differentiation
	immune dysregulation: ↑ levels of autoreactive antibodies

**Table 2 ijms-25-02259-t002:** The list of identified changes with pathogenicity or clinical significance that cannot be ruled out (likely pathogenic).

No.	Gene	Mutation by HGVS	#rs	Disorders	Frequency	Genotype
1	*C8A*	NM_000562.2:c.1228G>A NP_000553.1:p.AIa410Thr Chr1:57373634G>A Clinical significance: not reported in ClinVar.	rs767497299	Immunodeficiency due to late complement component deficiency [AR]	0.0008%	heterozygosity
2	*CR2*	NM_001006658.2:c.920C>T NP_001005659.1:p.Pro307Leu Chr1:207643142C>T Clinical significance: not reported in ClinVar.	rs138199106	Systemic lupus erythematosus, susceptibility to, 9, 610,927 [?], CVID, idiopathic immunoglobulin deficiency, primary hypogammaglobulinemia, common variable immunodeficiency, granulomatous inflammation associated with primary immunodeficiency [AR]	0.01%	heterozygosity
3	*HYOU1*	NM_001130991.1:c.579T>C NP_001130991.1:c.579T>C(p.=) Chr11:118925305A>G Clinical significance: not reported in ClinVar.	new	HYOU1 deficiency, congenial neutropenias IUISS2018 [AR]	?	heterozygosity
4	*LCK*	NM_001042771.1:c.601C>A NP_001036236.1:p.Gly201Ser Chr1:32741634G>A Clinical significance: in ClinVar, the change is described as VUS, one report.	rs11567841	Immunodeficiency 22, 615,758 [AR]	0.04%	heterozygosity
5	*NOD2*	NM_001293557.1:c.2938dup NP_001230486.1:p.Leu980ProfTer2 Chr16:50763778G>GC Clinical significance: according to ClinVar, the lesion is defined as a risk factor for the development of the following diseases: Yao syndrome, Blau syndrome, Crohn disease, Inflammatory Bowel Disease 1 (Crohn disease).	rs2066847	Sarcoidosis, early-onset, 609,464 [Mu]; childhood granulomatous arthritis; recurrent myositis; pediatric granulomatous arthritis, PGA; early-onset sarcoidosis; Blau syndrome [AD]; psoriatic arthritis, susceptibility to, 607,507 [?]; inflammatory bowel disease [Mu]	0.99%	heterozygosity
6	*C9*	NM_001737.3:c.499C>T NP_001728.1:p.Pro167Ser Chr5:39331894G>A Clinical significance: in ClinVar and in HGMD, the change is described as a polymorphism associated with a higher risk of age-related macular degeneration.	rs34882957	Immunodeficiency due to deficits in late components of the dopamine system [?, AR]; macular degeneration, age-related, 15, susceptibility to, 615,591 [?]	0.32%	heterozygosity
7	*DNMT3B*	NM_001207055.1:c.1191+60C>T Chr20:31384735C>T Clinical significance: not reported in ClinVar.	rs192769774	Immunodeficiency–centromeric instability–facial anomalies syndrome 1, 242,860 [AR]	0.06%	heterozygosity
8	*DOCK8*	NM_001193536.1:c.624G>A NM_001193536.1:c.624 G>A (p.=) Chr9:325671G>A	rs1156965453	CID due to DOCKS deficiency, DOCKS immunodeficiency syndrome, complex immunodeficiency due to DOCKS deficiency [AR]	0.0008%	heterozygosity
9	*UNC13D*	NM__99242.2:c.2335G>A NP_954712.1:p.Val779Met Chr17:73830188C>T Clinical significance: In ClinVar, the lesion is described as benign and VUS.	rs113861754	Familial HLH, familial hemophagocytic lymphohistiocytosis [AR?]	0.12%	heterozygosity

**Table 3 ijms-25-02259-t003:** Disease course.

Age (Years)	7	10	11–12	13	14	15	16
symptoms	first symptoms of arthritis	polyarthritis	leukopenia, neutropenia	chronic cough, dyspnea, skin lesions	salivary gland inflammation	leukopenia, neutropenia	right knee arthritis
tests (selected)	ESR N, CRP N, blood count N	ESR N; CRP N; blood count N; ANA 1:320; RF (-); IgG, IgM, IgE, IgA (N); myelogram N	blood count—leukopenia, neutropenia; myelogram N; anti-granulocyte antibodies (-); lymphocyte subpopulations: NK cells < N; IgG subclasses N;	chest X-ray N; spirometry—obstructive airway disease chest; CT N	ESR N, CRP N, blood count N, ANA 1:640, ENA (-), RF (-), hist-pat of the salivary gland—features of Sjogren’s disease	WES—de novo mutation in the *DOCK8* gene	MRI joint effusion, expansion of the synovium, bone erosion; hist-pat.
diagnosis	arthralgia	JIA	JIA adverse reactions, MTX suspected; drug allergy after the 2nd dose of ADA	Asthma Acanthosis nigricans Pityriasis rosea	JIA Sjogren’s disease asthma	*DOCK8* deficiency	PVNS
treatment	physiotherapy	MTX, SSA, GC	ADA; CS, HCH, GC, ETA	combination inhaler (GCs+bronchodilatator)	CS, GC, ETA; combination inhaler	CS, GC, ETA; combination inhaler	arthroscopic synovectomy

ADA—adalimumab; ANA—antinuclear antibody; CS—cyclosporin; CT—computed tomography; ENA—extractable nuclear antigen test; GC—corticosteroid; ETA—etanercept; HCH—hydroxychloroquine; JIA—juvenile idiopathic arthritis; N—normal; MTX—methotrexate, SSA—sulfasalazine; MRI—magnetic resonance imaging; WES—whole-exome sequence; PVNS—pigmented villonodular synovitis.

## Data Availability

Due to privacy and ethical concerns, the data that support the findings of this study are available on request from the First Author, (V.O.-W.).

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
