# Peer review of "DOCK8 Mutation in Patient with Juvenile Idiopathic Arthritis and Sjögren’s Syndrome"

_ijms, 2024, doi:10.3390/ijms25042259_

Round 1

Reviewer 1 Report

Comments and Suggestions for Authors

Major Comments:

Comment 1. The abstract should be more focused with necessary findings. The abstract lacks a clear statement of the study's objective and hypothesis.

Comment 2. The introduction provides a good overview, but it lacks a clear hypothesis or research question. Authors ensure to provide more recent references especially if there have been updates in the understanding of DOCK8 deficiency.

Comment 3.The case description is detailed and there is a need for better organization. Authors should consider dividing it into subsections such as "Clinical Manifestations," "Diagnostic Journey," and "Treatment History" for improved readability. Authors are suggested to clarify the timeline of events to make it easier for readers to follow the patient's medical history.

Comment 4. Authors should provide more information on the methodology of the Whole Exome Sequence studies. Also provide the supplementary material about the various reported results. The significance of the detected changes, especially in the BANK1 and NOD2 genes, needs more discussion. Authors are suggested to elaborate on the potential correlation with autoinflammatory diseases.

Comment 5. Authors are suggested to improve the discussion by comparing the presented case with existing literature. The discussion should explicitly address the novelty of the case, particularly in the context of JIA and Sjogren's disease and how it contributes to the existing knowledge.

Comment 6. Authors should provide the limitations of the study.

Comment 7. Authors should add practical implications of the study's findings for clinical practice and potential future research directions.

General Comments:

1. The manuscript needs thorough proofreading and editing for language and style. Some sentences are complex and may be challenging for readers. Simplify language where possible for better comprehension.

2. Authors should ensure that all references cited in the manuscript are included in the reference list, and vice versa.

Comments on the Quality of English Language

The manuscript needs thorough proofreading and editing for language and style. Some sentences are complex and may be challenging for readers. Simplify language where possible for better comprehension.

Author Response

Dear Reviewer,

We sincerely thank you for undertaking the review of our article “DOCK8 mutation in patient with juvenile idiopathic arthritis and Sjögren's syndrome”. Thank you for your informative and detailed review of our article. We believe that your excellent knowledge and commitment influenced our article and made it much better. We have followed your suggestions and all changes have been highlighted or highlighted in the updated version of the manuscript. Here are the point-by-point responses.

Reviewer 1

  1. The abstract should be more focused with necessary findings. The abstract lacks a clear statement of the study's objective and hypothesis.

RE: In accordance with the comments, the abstract has been revised and supplemented with missing elements. we hope that in this form it will meet expectations.

  1. The introduction provides a good overview, but it lacks a clear hypothesis or research question. Authors ensure to provide more recent references especially if there have been updates in the understanding of DOCK8

RE: Thank you for your constructive feedback. The research hypothesis is missing due to the nature of the clinical case. We also did not find more recent literature data than the ones we cited. However, the introduction has been gently restructured to strengthen the context.

  1. The case description is detailed and there is a need for better organization. Authors should consider dividing it into subsections such as "Clinical Manifestations," "Diagnostic Journey," and "Treatment History" for improved readability. Authors are suggested to clarify the timeline of events to make it easier for readers to follow the patient's medical history.

RE: As suggested, the chapter describing the case was redrafted by adding suggested subsections.

  1. Authors should provide more information on the methodology of the Whole Exome Sequence studies. Also provide the supplementary material about the various reported results. The significance of the detected changes, especially in the BANK1 and NOD2 genes, needs more discussion. Authors are suggested to elaborate on the potential correlation with autoinflammatory diseases.

RE: Thank you for your comment. We provided information about additional test which allowed to exclude any deletions in the BLANK1. Information was added.

  1. Authors are suggested to improve the discussion by comparing the presented case with existing literature. The discussion should explicitly address the novelty of the case, particularly in the context of JIA and Sjogren's disease and how it contributes to the existing knowledge.

RE: Thank you for your insightful comment. Our case has been discussed based on the available scientific literature. Precisely because of the novelty of the case, it is difficult to lead the discussion with more reports.

  1. Authors should provide the limitations of the study.

RE: The limitation of the study is the lack of whole-genome sequencing (WGS), which would enable the examination of non-coding sequences and determine the pathogenicity of the detected variant in the DOCK8 gene.

  1. Authors should add practical implications of the study's findings for clinical practice and potential future research directions.

RE: Thank you for your important comment. We have added in the conclusion a brief summary of how our reported case may interact with clinical practice.

Again, we would like to thank you for the effort and time and we are hoping that our manuscript in its current for will fulfill the requirements of the of the International Journal of Molecular Sciences.

Thank you for your time and consideration,

Reviewer 2 Report

Comments and Suggestions for Authors

In this Case Report, Opoka-Winiarska et al. provide a summary of a very informative case of DOCK8 deficiency: the clinical history of the patient represents a sort of "map" for the potential manifestation of this condition, that is shared with potential readers through an accurate reporting.

From my point of view, the present paper could be improved by the following interventions:

1) rows 202 and following could be anticipated in the earlier sections in order to improve the background reporting of the disease you're reporting on;

2) row 53-54: Authors referenced Table 1, but the content is more consistent with Table 3; Table 1 is not very informative and could be moved as supplementary material in order to summarize the care hereby reported;

3) Table 2 could be in turn moved to supplementary material;

4) Table 3 is very informative, but should be moved as Table 1

5) Table 4 could be improved by providing prevalence estimates from literature

Author Response

Dear Reviewer,

We sincerely thank you for undertaking the review of our article “DOCK8 mutation in patient with juvenile idiopathic arthritis and Sjögren's syndrome”. Thank you for your informative and detailed review of our article. We believe that your excellent knowledge and commitment influenced our article and made it much better. We have followed your suggestions and all changes have been highlighted or highlighted in the updated version of the manuscript. Here are the point-by-point responses.

Reviewer 2

In this Case Report, Opoka-Winiarska et al. provide a summary of a very informative case of DOCK8 deficiency: the clinical history of the patient represents a sort of "map" for the potential manifestation of this condition, that is shared with potential readers through an accurate reporting. From my point of view, the present paper could be improved by the following interventions:

  • rows 202 and following could be anticipated in the earlier sections in order to improve the background reporting of the disease you're reporting on;

RE: Thank you for your valuable suggestion. The chapter on case reporting has been rewritten in accordance with the recommendations of the second reviewer, so we hope that in its current form it will also meet this comment.

  • row 53-54: Authors referenced Table 1, but the content is more consistent with Table 3; Table 1 is not very informative and could be moved as supplementary material in order to summarize the care hereby reported;

RE: Thank you very much for your vigilance and rightful comment. At the stage of “gluing” the publication together there must have been a slight mistake in the numbering of the tables in the text. The tables have been replaced.

  • Table 2 could be in turn moved to supplementary material;

RE: As commented, the table has been moved to supplementary material.

  • Table 3 is very informative, but should be moved as Table 1

RE: The tables have been swapped places.

  • Table 4 could be improved by providing prevalence estimates from literature

RE: Unfortunately, there is no adequate data to supplement the frequency of symptoms. One of the reasons for describing the case, is precisely the paucity of literature on this particular disease. The table has been moved to supplementary material.

Again, we would like to thank you for the effort and time and we are hoping that our manuscript in its current for will fulfill the requirements of the of the International Journal of Molecular Sciences.

Thank you for your time and consideration,